# Typical 2-Cys Peroxiredoxins as a Defense Mechanism against Metal-Induced Oxidative Stress in the Solitary Ascidian *Ciona robusta*

**DOI:** 10.3390/antiox11010093

**Published:** 2021-12-30

**Authors:** Laura Drago, Diana Ferro, Rigers Bakiu, Loriano Ballarin, Gianfranco Santovito

**Affiliations:** 1Department of Biology, University of Padova, 35131 Padova, Italy; laura.drago@phd.unipd.it; 2Children’s Mercy Research Institute, Hospital and Clinics, Kansas City, MO 64108, USA; dferro@cmh.edu; 3Department of Pediatrics, University of Missouri-Kansas City, Kansas City, MO 64108, USA; 4Department of Aquaculture and Fisheries, Agricultural University of Tirana, 1000 Tiranë, Albania; rigers.bakiu@ubt.edu.al

**Keywords:** *Ciona robusta*, tunicate, metals, typical 2-Cys peroxiredoxins, antioxidant enzymes

## Abstract

Typical 2-Cys peroxiredoxins (2-Cys Prdxs) are proteins with antioxidant properties belonging to the thioredoxin peroxidase family. With their peroxidase activity, they contribute to the homeostatic control of reactive oxygen species (ROS) and, therefore, participate in various physiological functions, such as cell proliferation, differentiation, and apoptosis. Although Prdxs have been shown to be potential biomarkers for monitoring aquatic environments, minimal scientific attention has been devoted to describing their molecular architecture and function in marine invertebrates. Our study aims to clarify the protective role against stress induced by exposure to metals (Cu, Zn, and Cd) of three Prdxs (Prdx2, Prdx3, and Prdx4) in the solitary ascidian *Ciona robusta*, an invertebrate chordate. Here, we report a detailed pre- and post-translational regulation of the three Prdx isoforms. Data on intestinal mRNA expression, provided by qRT-PCR analyses, show a generalized increase for Prdx2, -3, and -4, which is correlated to metal accumulation. Furthermore, the increase in tissue enzyme activity observed after Zn exposure is slower than that observed with Cu and Cd. The obtained results increase our knowledge of the evolution of anti-stress proteins in invertebrates and emphasize the importance of the synthesis of Prdxs as an efficient way to face adverse environmental conditions.

## 1. Introduction

Industrial and mining activities, including the unwise use of pesticides, are considered the primary pollution sources of aquatic bodies. The abiotic factors, including xenobiotics, temperature, and hypo-osmotic stress, have been shown to affect the normal physiology of marine animals and induce physiological responses aimed at maintaining homeostasis [1]. These modifications require the regulation of the expression of stress response genes, such as those encoding antioxidant proteins [2,3,4,5,6,7,8], and among them, peroxiredoxins (Prdxs) [9,10].

Prdxs are thiol-specific antioxidant enzymes that have recently received significant attention for their role in stress responses, characterized by an imbalance between the reactive oxygen species (ROS) production rate and their removal from the cell. These proteins can modulate various physiological functions, such as cell proliferation, differentiation, and apoptosis, by decreasing peroxide levels and the risk of ROS-related damages [11,12]. With their peroxidase activity, Prdxs act to reduce and detoxify peroxynitrites (ONOO-), hydrogen peroxide (H_2_O_2_), and a wide range of different organic hydroperoxides (ROOH) by using their cysteine residues for catalysis [13,14,15].

Prdxs are broadly distributed among living organisms, from bacteria to mammals [16,17,18,19,20,21]. Although they are predominantly located in the cytosol, they have also been found in mitochondria, chloroplasts, and peroxisomes. The first Prdx was discovered in 1987, in yeast, and was named thiol-specific antioxidant (TSA) since it was thought to be able to remove reactive sulfur derivatives (e.g., •RS, •RSSR^−^, and RSOOH) rather than ROS. Subsequently, when TSA was shown to reduce peroxides by using thioredoxin (Trx) as an immediate hydrogen donor, the protein was renamed thioredoxin peroxidase (TPX). Lastly, it was renamed Prdx after realizing that some enzymes of this family do not rely on Trx as electron donors [22].

In mammals, six Prdx isoforms have been identified. They are classified into three subgroups: typical 2-Cys, atypical 2-Cys and 1-Cys [15]. All the Prdxs share the same catalytic mechanism in which an active-site cysteine (the peroxidatic cysteine, C_P_), containing a sulfur atom (C_P_-SH), confers the catalytic activity and is oxidized to a sulphenic acid (C_P_-SOH) by the peroxide substrate. The recycling of the sulphenic acid back to a thiol is what distinguishes the three enzyme subgroups in which C_P_ is highly conserved, whereas the “resolving” Cys (C_R_) residue can have different positions (2-Cys subgroups) or be absent (1-Cys subgroup) [23,24].

All the Prdx subgroups share a typical core organization containing five α-helices and seven β-strands [25]. In the reduced form (SH), the peroxidatic cysteine is in a solvent-accessible pocket, formed by a loop-helix structural motif, surrounded by three residues conserved in all the subgroups: Pro^44^, Thr^48^ and Arg^127^. The atypical 1-Cys Prdx has an additional active site, where, besides a histidine (His^26^) and an aspartic acid (Asp^140^) residue, a serine (Ser^32^) is present that allows the phospholipase activity [26]. Prdx1–5 belong to the 2-Cys subgroups, containing two conserved cysteines (C_P_ and C_R_). Prdx1–4 are included in the typical 2-Cys subgroup. They function as obligate homodimers since they form inter-subunit disulfide bonds between C_P_-SOH of a subunit and C_R_-SH of another one, reduced by cell-specific disulfide oxidoreductases to complete the catalytic cycle. Prdx5, included in the atypical 2-Cys subgroup, functions as a monomer, undergoing intra-subunit disulfide bonds between C_P_-SOH and C_R_-SH of the same subunit. The reduction in the disulfide bond is operated by Trx [27]. Lastly, Prdx6, included in the 1-Cys subgroup, forms homo- or hetero-dimers. Since it has only one highly conserved cysteine residue, C_P_-SOH, it is reduced by thiol-containing electron donor molecules (e.g., glutathione). In general, Prdx levels can be regulated by changes in the phosphorylation, redox and oligomerization states [15].

The measure of intracellular Prdx levels can be used in the biomonitoring of the aquatic environment as a critical indicator of pollution [28]. However, only a few Prdxs have been studied in aquatic invertebrates, mostly in crustaceans and mollusks [29,30], and the modulation of their synthesis by various pollutants needs to be explored. Therefore, this study aims to increase our knowledge of ROS scavenging in invertebrates exposed to high metal ion concentrations, using *Ciona robusta* as a reference organism.

This marine invertebrate is considered a reliable animal model for evolutionary studies, thanks to its peculiar phylogenetic position of invertebrate chordate, closely related to vertebrates [31]. It is also functional to ecotoxicological studies, as it is a filter-feeding organism able to bioaccumulate xenobiotics, even at low concentrations [32]. In the tunicate database ANISEED [33], we identified and characterized three transcripts for Prdx2, Prdx3, and Prdx4, respectively. Their genetic expression was analyzed at transcriptional and post-transcriptional levels in the intestine of *C. robusta* upon the exposure of animals to copper (Cu), zinc (Zn), and cadmium (Cd) at the same experimental conditions used in previous studies. Finally, we analyzed possible correlation of our results with those previously obtained with other anti-stress proteins, such as metallothioneins [34], the enzymes involved in glutathione (GSH) biosynthesis [35], phytochelatin synthase [36], Cu/Zn superoxide dismutase (Cu/Zn SOD) [37], glutathione peroxidase 7 [38] and proteins for stress granules nucleation [39].

## 2. Materials and Methods

### 2.1. Animals and Treatments

Specimens of *C. robusta* were sampled in Chioggia, near the Marine Station of the University of Padova, and transferred to the Department of Biology, University of Padova, where they were maintained in aerated aquaria, filled with filtered (0.45 μm) seawater (FSW). They were fed with Phyto Marine (Oceanlife, Bologna, Italy) and Reefpearls (D. van Houten, Groningen, the Netherlands) and acclimated for five days at 17 °C, alternating periods of light and darkness (12 h/12 h). Twenty-four animals were considered for each metal exposure and reared in separate aquaria. Storage solutions of CuCl_2_, ZnCl_2_, and CdCl_2_ were previously prepared in distilled water, and their concentrations were determined with a Perkin Elmer 4000 Atomic Absorption Spectrophotometer (Perkin Elmer, Waltham, MA, USA). The solutions were used to prepare working dilutions with a final sub-lethal concentration of 10 μM in FSW. One aquarium filled only with FSW hosted 24 animals used as reference controls. After 6, 24, 48, and 72 h of exposure, six specimens were collected from each aquarium, dissected, and their intestines were frozen in liquid nitrogen and stored at −80 °C until use.

### 2.2. Primers Design, Total RNA Extraction, cDNA Synthesis, Amplification, and Sequencing

In the ANISEED database (https://www.aniseed.cnrs.fr/ (accessed on 23 September 2021)), gene sequences related to *C. robusta* Prdxs are present. Their identities were validated by BLAST comparison (https://blast.ncbi.nlm.nih.gov/Blast.cgi (accessed on 23 September 2021)) to identify those of Prdx2, Prdx3, and Prdx4. Primers for PCR amplification (Appendix A) were designed in the coding region of putative Prdx2 (transcript ID: KY2019:KY.Chr8.1384.v1.nonSL16-1), Prdx3 (transcript ID: KY2019:KY.Chr10.1364.v1.ND1-1), Prdx4 (transcript ID: KY2019:KY.Chr10.1036.v1.SL1-5) and β-actin (transcript ID: KY2019:KY.Chr11.686.v1.nonSL30-1), the latter used as endogenous control. They were analyzed with the IDT Oligo Analyzer tool (https://eu.idtdna.com/pages/tools/oligoanalyzer (accessed on 23 September 2021)) and synthesized by BMR genomics SRL (Padova, Italy).

Total RNA was extracted from the intestines of *C. robusta* by precipitation, using Biozol (Hangzhou Bioer Technology, Hangzhou, China) as the lysis buffer according to the product manual specifications. RNA concentration and purity were assessed by the A_260/280_ and A_260/230_ ratios, with a Nanodrop ND-1000 spectrophotometer (ThermoFisher Scientific, Waltham, MA, USA). Its integrity was proved by the visualization of rRNAs in 1% agarose gel, using Midori Green Direct (Nippon Genetics, Tokyo, Japan) as an intercalating agent and loading dye.

The reverse transcription was performed on 1 μg of total RNA, using the ImProm-II™ Reverse Transcription System (Promega, Madison, WI, USA) kit (Oligo-dT Anchor primer is reported in Appendix A), as previously described [38]. PCR amplifications were carried out with GRS Taq DNA polymerase (5 U/μL; Grisp, Porto, Portugal), according to the following cycling parameters: 95 °C for 2 min, then 40 cycles at 95 °C for 30 s, melting temperature (Tm) (Appendix A) for 30 s, 72 °C for 1 min, and the last step at 72 °C for 10 min. PCR amplicons were visualized on 1.5% agarose gel with Midori Green Direct (Nippon Genetics) and purified with the Wizard^®^ SV Gel and PCR Clean-Up System (Promega) kit. The transcription in the intestine of *C. robusta* for Prdx2, Prdx3, and Prdx4, was confirmed by Sanger sequencing of the PCR amplicons, performed by BMR genomics, on the Applied Biosystems 3730XL 96-capillary DNA Analyzer (Applied Biosystems, Foster City, CA, USA). The sequencing results allowed us to design specific primers for quantitative real-time PCR (qRT-PCR) (Appendix A).

### 2.3. Quantitative Real-Time PCR (qRT-PCR)

The transcription levels of Prdx isoform 2, 3, and 4, in metal-exposed animals and control specimens were analyzed by qRT-PCR, using the qPCRBIO SyGreen Mix Separate-ROX kit (PCR Biosystems, Wayne, PA, USA). For each treatment, samples of cDNA from six different animals were considered as biological replicates, and each sample was run three times (technical triplicate). The qRT-PCR primers for the three Prdx isoforms and the housekeeping gene (*β-actin*) are reported in Appendix A. After a preliminary validation of the primer amplification efficiency (absolute quantification), qRT-PCR was performed on 20 μL of SYBR Green-ROX mix, containing 50 ng of cDNA, using the Applied Biosystems 7900-HT Fast Real-Time PCR System with the following cycling parameters: 95 °C for 2 min (denaturation), then 45 cycles at 95 °C for 20 s and 60 °C for 1 min (annealing), and the last phase at 95 °C for 15 s, 60 °C for 1 min, 95 °C for 15 s and 60 °C for 15 s (extension). The dissociation curve confirmed the absence of genomic contamination. Relative quantification values were obtained using the Pfaffl mathematical model (2^−ΔΔCt^ calculation) [40]. The amounts of expression level were normalized with respect to the housekeeping gene as in previous studies [37,38].

### 2.4. Tissue Activity of 2-Cys Prdxs

To assess the Prdx2, Prdx3, and Prdx4 activities, intestines from treated and non-treated specimens were homogenized according to the 2-Cys Peroxiredoxin Activity Assay protocol (Redoxica, Little Rock, AR, USA). In addition, cell extracts were used to evaluate the protein concentration, enzymatic activity, and metal content.

The protein concentration in cell extracts was measured with the Lowry method, based on the colorimetric reaction with the Folin–Ciocalteu reagent, sensitive to protein quantities of 5–100 µg [41]. Protein quantification was performed with a PerkinElmer Lambda EZ-201 UV/VIS spectrophotometer at 750 nm.

To determine the 2-Cys Prdx activity in cell extracts, two reaction solutions were prepared, with and without Trx, to determine the disappearance of NADPH caused by the reaction catalyzed by thioredoxin reductase. The decrease in NADPH concentration was measured at 340 nm with Agilent 8453 UV-visible spectrophotometer (Agilent, Santa Clara, CA, USA) for 120 s after adding H_2_O_2_ as the substrate. The 2-Cys Prdx activity was evaluated as a micromole of oxidated NADPH per min at 25 °C. Data were then normalized against the total protein concentration, and bovine serum albumin was used as the standard.

### 2.5. Analyses of Tissue Metal Content

The Cu, Zn and Cd contents in the cell extracts were determined by atomic absorption spectrophotometry. The instrument for metal analyses (PerkinElmer mod. 4000) was calibrated by the standard addition method and new standard salt solutions. The concentration obtained was normalized with data of the total protein concentration in the same samples.

### 2.6. Statistical Analyses

Data are expressed as the mean of six different biological samples (n = 6) ± standard deviation. Statistical analyses were performed with the PRIMER statistical program, using the ANOVA test followed by Duncan’s test, to evaluate significant differences between treated and untreated samples (*p* < 0.05).

### 2.7. Gene and Protein Organization Analyses

Prdx nucleotide sequences for the isoforms 2, 3 and 4 of *C. robusta*, obtained by amplicon sequencing, were completed in silico, thanks to the ANISEED database, and then translated with the ExPASy translate tool (https://web.expasy.org/translate/ (accessed on 23 September 2021)). Exon/intron composition was analyzed by matching the cDNA with the genomic sequence found in the ANISEED database, using the Splign tool (https://www.ncbi.nlm.nih.gov/sutils/splign/splign.cgi (accessed on 23 September 2021)). Two thousand nt upstream of the transcript sequence were analyzed with the Primer Premier 5.00 software package (Primer Biosoft International, Palo Alto, CA, USA) to identify putative transcription factor binding sites, such as antioxidant response element (ARE), xenobiotic response element (XRE) and metal response element (MRE). The 3′-UTR regions were analyzed in silico to find putative sequences of rapid degradation and polyadenylation.

The domain composition of Prdxs was predicted with the SMART program (http://smart.embl-heidelberg.de/smart/set_mode.cgi?NORMAL=1 (accessed on 23 September 2021)). On the alignment, obtained with Clustal Omega (https://www.ebi.ac.uk/Tools/msa/clustalo/ (accessed on 23 September 2021)), comparing the amino acid sequences of *C. robusta* and orthologous sequences of other metazoans collected from GenBank (https://www.ncbi.nlm.nih.gov/nucleotide/ (accessed on 23 September 2021)), the domain architecture was visualized. The LALIGN tool (https://www.ebi.ac.uk/Tools/psa/lalign/ (accessed on 23 September 2021)) was used to compare the amino acid sequences of Prdxs two by two, and the NCBI CD-search tool (https://www.ncbi.nlm.nih.gov/Structure/cdd/wrpsb.cgi (accessed on 23 September 2021)) was used to classify them correctly.

### 2.8. Sequence Alignments and Phylogenetic Analyses

The cDNA and amino acid sequences of Prdx2, Prdx3, and Prdx4 of *C. robusta* were used for multiple alignment analyses with the T-Coffee package (Comparative Bioinformatics Group, Barcelona, Spain) [42], together with orthologous sequences of other metazoans, available in GenBank and ANISEED databases (Appendix A). T-Coffee was used to align Prdx2, Prdx3 and Prdx4 sequences. Even though the method is based on the popular progressive approach to multiple alignments, it is characterized by a dramatic improvement in accuracy with a modest sacrifice in speed, compared to the most used alternatives.

jModelTest 2.0 [43] was used to select the best-fit model of nucleotide substitution systematically. Analyses were performed using 88 candidate models and three types of criteria (Akaike information criterion (AIC), corrected Akaike information criterion (cAIC), and Bayesian information criterion (BIC)). Finally, ProtTest 3 [44] was used to select the best-fit model to analyze protein evolution. One hundred and twenty-two candidate models and the three previously mentioned criteria were used in these statistical analyses.

Phylogenetic trees were built using the Bayesian inference (BI) method implemented in Mr. Bayes 3.2 [45]. Four independent runs, each with four simultaneous Markov Chain Monte Carlo (MCMC) chains, were performed for 1,000,000 generations sampled every 1000 generations. Furthermore, we also used the maximum likelihood (ML) method implemented in PhyML 3.0 [46]. Bootstrap analyses were performed on 100,000 trees using nearest neighbor interchange (NNI) as a method for improvement of tree topology. FigTree v1.4.4 software (GitHub, San Francisco, CA, USA; http://tree.bio.ed.ac.uk/software/figtree/ (accessed on 23 September 2021)) was used to display the annotated phylogenetic trees.

### 2.9. In Situ Hybridization (ISH)

Amplicons obtained with primers designed for the qRT-PCR (Appendix A) were ligated in pGEM T-Easy Vector (Promega) and cloned in XL1-Blue Escherichia coli cells (Invitrogen, Thermo Fisher Scientific, Waltham, MA, USA). UltraPrep (AHN Biotechnologie, Nordhausen, Germany) kit was used to extract plasmid DNA from positively screened colonies, which was then sequenced by Eurofins Genomics (Eurofins Genomics Ebersberg, Germany) to check the successful ligation of the insert. Sense and anti-sense riboprobes were synthesized with the SP6 Polymerase (Promega) kit and purified with mini-Quick Spin Columns™ (Roche Molecular Systems, F. Hoffmann-La Roche AG, Basel, Switzerland).

As carried out in previous studies, ISH was performed on hemolymph extracted from animals exposed for 3 days to metal (FSW in controls) [39]. After pelleting the cells by centrifuging (800× *g* for 10 min), hemocytes were resuspended in FSW to a concentration of 10^5^ cells/mL and left to adhere on Superfrost Plus glass slides (Menzel-Glaser, Thermo Fisher Scientific, Waltham, MA, USA) for 30 min. After fixation with a freshly prepared solution of 4% paraformaldehyde, 0.1% glutaraldehyde in 0.2 M cacodylate buffer, pH 7.4, for 30 min, cells were washed in phosphate-buffered saline (PBS: 1.37 M NaCl, 0.03 M KCl, 0.015 M KH_2_PO_4_, 0.065 M Na_2_HPO_4_, pH 7.2), permeabilized in 0.1% Triton-X in PBS for 5 min and washed again in PBS. The prehybridization step was performed with the Hybridization Cocktail 50% Formamide (Amresco, VWR International, Radnor, PA, USA) for 1 h at 58 °C, followed by hybridization at the same temperature in the same solution with 2 μg/mL of biotin-labeled riboprobe overnight. Next, cells were washed in SSC (0.3 M NaCl, 40 mM sodium citrate, pH 4.5), then three times in 50% formamide in SSC for 30 min each at 58 °C and finally in PBS containing 0.1% Tween-20 (PBS-T). Samples were incubated in the dark in a solution of 5% H_2_O_2_ in methanol for 30 min, washed with PBS-T, and treated with Vectastain ABC (Vector Laboratories, Burlingame, CA, USA) for 30 min. Cells were stained with 0.025% 3,3′-diaminobenzidine (DAB; Sigma-Aldrich, Merck, Darmstadt, Germany) and 0.004% H_2_O_2_ in PBS for 15 min in the dark, dehydrated with ethanol, and mounted with Eukitt (Electron Microscopy Sciences, Hatfield, PA, USA).

## 3. Results

### 3.1. Gene and Transcript Organizations

The sequencing of the amplicon obtained with CrPrdx2fw-CrPrdx2rv primers on a cDNA from *C. robusta* intestine returned a partial sequence of 478 nt, belonging to the putative Prdx2 nucleotide sequence of *C. robusta*, previously collected from the ANISEED database and used to design the primers reported above. This result suggests that the gene, named *cr-Prdx2* and present in chromosome 8, is transcribed in *C. robusta*. The in silico analysis indicated that the whole transcript is 667 nt in length, with 5′- and 3′-UTR regions of 9 nt and 64 nt, respectively. The open reading frame (ORF) is 594 nt long and encodes a putative protein of 197 aa, with a deduced molecular weight of 21.81 kDa. The gene is organized in four exons and three introns, with the ATG start codon located in the first exon, 114 nt long, and the TAA stop codon located in the last one, 278 nt long. In the 3′-UTR region, two nucleotide sequences, corresponding to rapid degradation (ATTTA) signals, are conserved at nt 635 and 641 (Appendix A). In the promoter region of cr-*Prdx2*, no regulatory sequences ascribable to ARE or MRE were found.

Regarding Prdx3, the amplicon obtained with CrPrdx3fw-CrPrdx3rv primers allowed us to identify a partial sequence of 631 nt included in the sequence found in ANISEED, thus confirming that the gene, named *cr-Prdx3* and located in chromosome 10, is transcribed in the intestine of *C. robusta*. The in silico analysis indicated that the whole transcript is 1190 nt in length and contains an ORF of 708 nt, flanked by 5′- and 3′-UTR regions of 120 nt and 362 nt, respectively. The ORF encodes a putative protein of 235 aa, with a deduced molecular weight of 26.10 kDa. The gene is organized in five exons, with the ATG start codon located in the first exon, 162 nt long, and the TAA stop codon located in the last one, 413 nt long. In the 3′-UTR, three conserved sequences correspond to polyadenylation (AATAAA) signals (nt 914, 967, and 1164) and six to rapid degradation (ATTTA) signals (nt 864, 892, 945, 976, 994, and 1001) are present (Appendix A). Furthermore, in the promoter region of *cr-Prdx3*, one putative ARE (conserved core TGACNNNGC), 1601 nt upstream the start codon, and one putative MRE (conserved core TGCRCNC), 451 nt upstream the start codon, were identified.

Sequencing of the amplicon obtained with CrPrdx4fw-CrPrdx4rv primers returned a partial sequence of 585 nt. The putative Prdx4 sequence from the ANISEED database contains the sequence we obtained with the amplicon sequencing, thus confirming that the transcription of the gene, named *cr-Prdx4* and localized in chromosome 10, occurs in the *C. robusta* intestine. The whole transcript is 1241 nt long, with an ORF of 720 nt, located between a 5′-UTR region of 60 nt in length and a 3′-UTR region of 461 nt in length. The putative protein contains 239 aa and has a deduced molecular weight of 26.83 kDa. The gene is organized in a single exon of 1240 nt. In the 3′-UTR, there is a conserved sequence corresponding to the polyadenylation (AATAAA) signal, at nt 1162, and three conserved rapid degradation (ATTTA) signals, at nt 972, 994, and 1125 (Appendix A). In addition, putative conserved transcriptional regulation motifs were found in the promoter region of *cr-Prdx4*: one putative ARE, 1514 nt upstream the start codon, and one putative MRE, 364 nt upstream the start codon.

### 3.2. Protein Organizations

The NCBI Conserved Domain Search tool revealed that all the three Cr-Prdx isoforms belong to the thioredoxin proxidase-like superfamily, to the typical 2-Cys Prdx subfamily (Cr-Prdx2 *E*-value 1.37 × 10^−117^; Cr-Prdx3 1.36 × 10^−117^; Cr-Prdx4 2.70 × 10^−117^). Orthologous amino acid sequences of invertebrates and vertebrates were considered for the multi-alignment analyses, obtained with Clustal Omega, for all the three Prdx isoforms, including *C. robusta* (Appendix A; transcript IDs are reported in Appendix A). Prdx2, -3 and -4 present two characteristic domains: the alkyl hydroperoxide reductase (AhpC)-thiol specific antioxidant (TSA) domain, where the AhpC domain is responsible for directly reducing organic hyperoxides and the TSA domain constitutes an enzymatic defense against sulfur-containing radicals; and the C-terminal 1-cysPrdx domain, crucial for providing the extra cysteine necessary for dimerization of the whole molecule. Loss of the peroxidase activity is associated with oxidation of the catalytic cysteine found upstream of this domain.

Regarding Prdx2, in the AhpC-TSA domain, 34.3% of amino acid residues are completely conserved among the examined species. In the C-terminal domain, 31.6% of the amino acids are completely conserved (Appendix A). In this alignment, the putative Cr-Prdx2 amino acid sequence shows the highest identity with those of *Danio rerio* and *Phyton bivittatus* (74.6%) and the highest similarity with that of *Mus musculus* (92.4%) (Appendix A), as revealed by the LALIGN tool.

The AhpC-TSA domain of Prdx3 shows 17.9% of conserved amino acids, whereas, in the C-terminal domain, they are 36.8% completely conserved (Appendix A). The putative Cr-Prdx3 amino acid sequence shows the highest identity (72.4%) with the *Xenopus laevis* orthologous sequence and the highest similarity (90.1%) with the *Drosophila melanogaster* orthologous sequence (Appendix A).

For Prdx4 (Appendix A), the AhpC-TSA domain results in 60.4% conservation among the species considered for multi-alignment, whereas the C-terminal domain shows 29.4% conservation. The putative Cr-Prdx4 amino acid sequence shows the highest identity (88.9%) and similarity (96.6%) with the *Botrylloides leachii* orthologous sequence (Appendix A).

In all the analyzed peroxiredoxins, the four amino acids responsible for the peroxidase activity were identified: proline (Pro_44_ in Cr-Prdx2, Pro_81_ in Cr-Prdx3, and Pro_79_ in Cr-Prdx3), threonine (Thr_48_ in Cr-Prdx2, Thr_85_ in Cr-Prdx3, and Thr_83_ in Cr-Prdx4), cysteine (Cys_51_ in Cr-Prdx2, Cys_88_ in Cr-Prdx3 and Cys_86_ in Cr-Prdx4) and arginine (Arg_127_ in Cr-Prdx2, Arg_164_ in Cr-Prdx3 and Arg_162_ in Cr-Prdx4) (Appendix A).

### 3.3. Molecular Phylogeny

jModelTest 2.0 software indicated that the GTR+I+G model is the best-fit model to analyze the evolution of the coding sequences of all the Prdx isoforms, with a gamma shape value (four rate categories) of 0.868 using all statistical criteria: AIC, cAIC and BIC (−lnL = 30,440.51). Figure 1 shows all the Prdx isoforms phylogenetic tree generated by applying the BI and ML methods to the data set of coding sequences.

The cladogram shows that two significant clusters, corresponding to Prdx3 and Prdx4, are well defined and separated from the Prdx2 sequences (posterior probability 98%, bootstrap value 50%). However, there are some exceptions, such as in *Clonorchis sinensis*, *Aplysia californica* and *Culex pipiens quinquefasciatus* Prdx2s that cluster within the Prdx3 clade and *Crassostrea gigas* Prdx2, which is included inside the Prdx4 clade. It is to note that Prdx2 of *B. leachii* emerges with the Prdx4 of this species (posterior probability 100%, bootstrap value 100%), and the Prdx4 sequences from other tunicates (*Ciona intestinalis* and *Ciona robusta*; posterior probability 100%, bootstrap value 88%). Conversely, within the Prdx3 clade, all tunicate sequences are clustered together with urinary blood fluke *Schistosoma haematobium* (posterior probability 78%). In Prdx2s, species of the genus *Ciona* are grouped in a clade (posterior probability 100%, bootstrap value 100%), well separated from the other Prdx2 sequences (posterior probability 100%).

ProtTest3 statistical results determined that the WAG+G model was the best model to apply for the phylogenetic analysis of Prdx amino acid sequences, with a gamma shape value (four rate categories) of 0.444 using all statistical criteria (−lnL = 13,169.13). Appendix A shows a Prdx phylogenetic tree generated by applying the BI and ML methods to the data set of amino acid sequences. The cladogram topology is quite similar to the previous one, though in this cladogram, three clades are identified, each including the sequences of a specific isoform, with the same exceptions previously highlighted (in this case, *Clonorchis sinensis* Prdx2 emerged close to Prdx4s).

### 3.4. qRT-PCR

The three genes, *cr-Prdx2*, *cr-Prdx3*, and *cr-Prdx4*, are transcribed in the *C. robusta* intestine, both in control conditions and after metal exposures. Their gene expression levels are reported in Figure 2.

The levels of Prdx2 mRNA measured in organisms that were exposed to Cu showed a time-dependent increase starting from 24 h (Figure 2a). At 72 h, they were about three times higher compared to the controls (*p* < 0.001). In the specimens exposed to Zn, the Prdx2 messenger levels were consistently higher than the controls (Figure 2b). The maximum expression was measured at 6 h to be almost three times higher than the controls (*p* < 0.01). At 24, 48 and 72 h, the expression values decreased slightly, remaining, however, doubled, compared to the controls (*p* < 0.01). Exposure to Cd also determined a statistically significant (*p* < 0.01) increase in the Prdx2 mRNA levels, compared to the controls for all the considered times (Figure 2c). The time-course of messenger accumulation was characterized by an initial overexpression followed by a down-regulation at 24 h, and a subsequent new increase at 48 h when the maximum mRNA expression for Prdx2 was recorded (about ten times compared to the controls). Subsequently, there was a further decrease, until the return to the levels measured at 6 h.

In the specimens of *C. robusta* exposed to Cu (Figure 2d), the levels of Prdx3 mRNA remained almost constant throughout the experiment, being approximately 50% greater than those measured in the controls only at 24 and 72 h (*p* < 0.05). The mRNA levels for Prdx3 were consistently higher than the controls in the organisms exposed to Zn for the first 48 h (Figure 2e). Maximum expression was measured at 6 h, with an increase in expression almost three times greater than the respective control (*p* < 0.001). At 24, 48 and 72 h, the expression values slightly decreased, remaining, however, higher than the controls. In organisms treated with Cd, there were no statistically significant differences compared to the controls except at 6 h (*p* < 0.01), where the level of Prdx3 mRNA was also statistically higher than at the other treatment times (Figure 2f).

Prdx4 mRNA levels measured in Cu-exposed organisms showed a statistically significant increase starting at 24 h (Figure 2g). At 72 h, they were about two times higher than the controls (*p* < 0.001). In organisms exposed to Zn, the messenger levels for Prdx4 remained consistently higher than the controls (*p* < 0.01) (Figure 2h). Exposure to Cd also resulted in a statistically significant (*p* < 0.01) increase compared to the controls of the Prdx4 messenger levels for all the times considered (Figure 2i). The time course of the accumulation of mRNA in the treated specimens showed an initial overexpression (about three times compared to the controls), which remained constant up to 48 h. Subsequently, there was a down-regulation, which resulted in values more similar to those measured in untreated organisms.

### 3.5. 2-Cys Prdx Activity

We measured the 2-Cys Prdx activity on cell-free extracts of *C. robusta* intestine. Exposure to Cu led to a statistically significant (*p* < 0.001) increase in the levels of 2-Cys Prdx active protein, for all the considered times (Figure 3). In particular, there was a regular increase up to 48 h, which was followed by a marked decrease at 72 h, when halved tissue activity was found.

In organisms exposed to Zn, the levels of active protein remained unchanged with respect to untreated organisms up to 24 h (Figure 3). Starting from 48 h, there was a progressive increase in tissue activity, at first three times higher than the controls (*p* < 0.001) and about five times higher (*p* < 0.001) at 72 h, when maximum expression was recorded.

For organisms exposed to Cd, statistically significant variations in the levels of 2-Cys Prdx active protein were found only at 6 and 48 h, with values, respectively, of three and six times higher than in the controls (*p* < 0.001). These peaks of expression were interspersed with downregulation events that brought 2-Cys Prdx activity back to the values measured in untreated specimens.

### 3.6. Metal Contents

The accumulation of Cu in organisms exposed to this metal is shown to be greater than values found in untreated organisms for all the times considered. At 6 and 24 h, the increase in Cu concentration is about three times (*p* < 0.01); at 48 h there is a further five times increase (*p* < 0.01); and finally, at 72 h, the maximum concentration value is well thirty times higher (*p* < 0.001) than that measured in the controls (Figure 4a).

Instead, in the cell-free extracts of organisms exposed to Zn, significant variations in concentrations of this metal emerged only at 6 and 48 h, with values three and two times higher (*p* < 0.001) than in the controls, respectively (Figure 4b). These peaks of accumulation were interspersed with times when the Zn concentration was quite similar to that measured in the untreated specimens.

In organisms exposed to Cd, there is a slight increase with respect to controls (*p* < 0.001) in the concentrations of this metal from 24 to 72 h (Figure 4c). The time course of the Cd accumulation was characterized by increasing values up to 48 h, when the maximum accumulation of this metal was observed. Subsequently, at 72 h, the tissue levels of this metal began to decrease.

### 3.7. ISH

For cr-*Prdx2* and cr-*Prdx3*, the ISH analysis on hemolymph revealed that granular amoebocytes were the only immunocytes expressing the Prdx mRNAs (Figure 5a,b). In the case of cr-*Prdx4*, besides granulocytes (Figure 5c), also phagocytes are active in the expression of the transcript (Figure 5d). No labeling was observed with the use of sense riboprobes.

## 4. Discussion

Marine organisms have adaptations that allow them to maintain homeostasis, ensuring growth and reproduction, in a wide variety of environmental conditions. Like any other metabolic pathway, the processes that lead to the production of ROS also vary significantly, depending on the variations of many environmental factors, and the regulation of antioxidant defenses is essential to keep the concentration of these molecules at sufficiently low levels in order to prevent oxidative stress and cell death. Many marine animals produce ROS in response to exposure to xenobiotics [47,48]. The production of ROS and the activation of various components of the antioxidant defense system, such as enzymes that catalyze the biosynthesis of glutathione, superoxide dismutase and glutathione peroxidase, have also been demonstrated in the solitary ascidian *Ciona* [35,37,38].

Despite a large amount of research on the antioxidant defense system of marine invertebrates, there is still much to discover concerning the molecular components and their mutual functional relationships. The present work characterized the genetic sequences of three Prdxs: *cr-Prdx2*, *cr-Prdx3*, and *cr-Prdx4*. Furthermore, we demonstrated, for the first time, that they are transcribed and expressed as active proteins in the intestine of *C. robusta*. Upon metal exposure, they increase their transcription, and an increase in the Prdx activity of cell extracts was also observed. Through the analysis of 5′ UTR, we identified putative regulatory sequences in *cr-Prdx3* and *cr-Prdx4*, i.e., ARE and MRE, the latter being of particular importance in the regulation of mRNA transcription in response to metals. This suggests that *C. robusta* has evolved a defense system against oxidative damages induced by metals.

The phylogenetic analyzes carried out on the nucleotide sequences of the three considered Prdx isoforms made it possible to add new information to the molecular evolution of anti-stress proteins. In particular, they confirmed a high level of nucleotide and amino acid conservation, with cladograms that are highly resolved, while maintaining an excellent homogeneity of sequences distribution within the representative clusters of the various taxa. An interesting result is that for none of the three isoforms, it was possible to identify the sequences of the tunicates as sister groups of the vertebrate sequences, unlike what was evidenced by recent phylogenetic studies [49]. However, this finding could be partially expected because analyses carried out on other gene sequences (γ-glutamyl-cysteine ligase, glutathione synthetase, superoxide dismutase, glutathione peroxidase, TIA-1 related nucleolysin, and tristetraproline) showed phylogenetic relationships similar to those obtained here [35,37,38,39]. In addition, there is the difficult determination of the phylogeny of tunicates and their peculiar intermediate position between vertebrates and invertebrates to be considered. In fact, it is not surprising that the positioning of tunicates as sister groups of vertebrates is the result of the most advanced and recent research works [31] and of years of heated scientific debate [50]. Another interesting result is represented by the peculiar position of *B. leachii* Prdx2, which emerges together with the isoform 4 of this species rather than its orthologous proteins. While maintaining the utmost caution in relation to the absence of other molecular data, we could hypothesize a gene duplication event that occurred relatively recently, following the differentiation between the progenitor of Prdx2 and that of Prdx4.

As regards the phylogenetic analyses performed on the amino acid sequences, cladograms were obtained, which show a practically identical topology to that of the cladograms obtained with the nucleotide sequences, despite having a lower degree of resolution. This result is in line with the high degree of conservation of amino acid sequences and could be the result of a strong purifying selection acting on them, an evolutionary feature common to other genes encoding antioxidant enzymes [16,17,19,51,52]. The action of purifying selection is important for the evolution of gene families, as it guarantees the maintenance of function of the encoded proteins. This is also clear from the multi-alignment analysis, which demonstrated that the amino acids, which are essential for their peroxidase activity, are highly conserved in the Prdx of *Ciona*, where the catalytic tetrad consisting of proline, threonine, cysteine and arginine plays an important role. A histidine, a serine and an aspartic acid are also important for this enzymatic activity.

The results obtained from the qRT-PCR and enzymatic activity analyses indicate that the genes encoding all the considered Prdx isoforms are expressed in the intestine of *C. robusta*. Of the three isoforms, the two that were most expressed at baseline are Cr-Prdx2 and Cr-Prdx4, being 1.5 times more expressed than Cr-Prdx3. The former is mainly expressed in the cytoplasm, and its hyperoxidation is believed to be physiologically relevant in relation to inflammatory and necrosis phenomena [53,54]. The second resides as a soluble protein in the lumen of the endoplasmic reticulum (ER), an intracellular structure that plays a fundamental role in many metabolic processes, including the biosynthesis of lipids and proteins, and in the detoxification processes [55]. It is known that the correct folding of the tertiary structure of proteins occurs in the ER with the formation of disulfide bridges. One of the most conserved enzymes that performs this function is the endoplasmic reticulum oxidoreductin 1 (ERO1), which, during its oxidative activity, produces H_2_O_2_ [56]. Considering this, it is not surprising that an isoform of Prdx is specifically expressed within the lumen of the RE, where it can exercise its protective function as a peroxide scavenger. Furthermore, it was confirmed that Prdx4 is also secreted [57], as a specific signal is present in its N-terminal portion, and therefore, its expression is also associated with inflammatory phenomena [58].

The metal sub-lethal exposures bring a generalized increase in gene expression levels for all the three Cr-Prdxs in the intestine of *C. robusta*, as revealed by qRT-PCR analyses. Cd seems to be the metal that primarily induces a generalized, statistically significant increase in the transcription levels, compared to controls of the Cr-Prdx isoforms, except Cr-Prdx3. This increase is also reflected at the post-transcriptional level, as detected in the total enzymatic activity of Prdx 2-Cys. Although we cannot discriminate the contribution of each Cr-Prdxs, we may suppose that this result refers especially to Cr-Prdx2, the mRNA accumulation time course of which is superimposable to that of the enzymatic activity, with the maximum peak at 48 h and minimum peaks at 24 and 72 h, when the amount of active proteins achieves the control levels. These two events may be justified by the overexpression of other components of the antioxidant system as demonstrated, under the same experimental conditions, for the enzymes catalyzing the GSH biosynthesis [35] and for Cu/Zn SOD [37]. We also observed that the accumulation of Cd in the intestine of *C. robusta* increases starting from 24 up to 48 h, when it reaches its maximum, and then decreases again. The reduction in the intracellular concentration of Cd may be related to its partial accumulation in the granular amoebocytes of the hemolymph, which then migrate into the tunic, the outer covering of tunicates, which becomes a non-systemic storage site, where these cells undergo cell death by apoptosis [36]. ISH on circulating hemocytes showed that the transcripts for *cr-Prdx2*, *cr-Prdx3*, and *cr-Prdx4* are mainly located in granular amoebocytes, as demonstrated in previous works [34,35,36,37,39], confirming their role as the circulating detoxification system [59]. For cr-Prdx4, granulocytes are not the only immunocytes involved in the detoxification processes from metals. In addition, phagocytes are involved. The role of phagocytes was demonstrated by Franchi et al. [60] in the colonial ascidian *Botryllus schlosseri*, which are active in the transcription of stress-related genes, such as glutathione synthase (GS), superoxide dismutase (SOD), and glutathione peroxidase 5.

Additionally, Cu induces an increase in the transcription for all the three Cr-Prdx isoforms, particularly Cr-Prdx2, even if in a less marked way than Cd. This was a partly unexpected result considering the redox properties of Cu, which catalyzes Haber–Weiss and Fenton reactions leading to hydroxyl radicals (•OH) formation among the most active ROS [61]. A consistent increase in the enzymatic activity in comparison to controls was revealed during the time course. We hypothesize that the mRNA for the three Cr-Prdx isoforms was readily translated into active protein when the intracellular concentration of Cu increased, also increasing the risk of oxidative stress. The metal accumulation reaches its maximum at 72 h, when we detect the maximum mRNA expression for Cr-Prdx2 and Cr-Prdx4. This physiological response seems justified by the fact that the Cu is an essential metal, so it can only damage the cell when its concentration increases beyond a threshold.

Exposure to Zn also has much less marked effects on the transcription of genes encoding Cr-Prdx isoforms 2, 3, and 4 than Cd. The maximum increase in mRNA expression levels is observed at 6 h, in concomitance with a peak of the metal accumulation. After that, a general decrease occurs, which, for Cr-Prdx3, marks the return to control levels. The time course of intracellular concentrations of Zn may be a phenomenon similar to that demonstrated for Cd, with partial accumulation in the granular amoebocytes and then extrusion into the tunic [36]. However, the accumulation of active protein begins at 48 h, reaching a maximum peak at 72 h, when the transcription levels for isoforms 2 and 4 seem to increase again. These data suggest a double type of time-dependent regulation on the accumulation of Zn in the *C. robusta* intestine: in the early exposure, there is a transcriptional regulation, so the transcription for Cr-Prdx 2, 3 and 4 occurs. In the later exposure, there is a post-transcriptional regulation, with the mRNAs translated into active proteins. Our results suggest that part of the transcripts might not immediately be translated, as observed in previous studies on components of the antioxidant system in animals exposed to oxidative stress conditions [17,51,52,62]. It is known that there are intracellular compartments, such as P-bodies and stress granules (SGs), in which mRNAs are stored, undergoing degradation or future translation, respectively [63,64,65]. Our previous study demonstrated the importance of SGs in regulating metal-induced stress responses in *C. robusta* [39]. SG nucleation occurs with the overexpression of mRNA binding proteins, such as TIA-1 related nucleolysin (TIAR) and tristetraprolin (TTP), able to specifically recognize ARE sequences in the promoters for some anti-stress proteins, such as Prdxs. The hypothesis is, therefore, that in the early phase of Zn exposure, mRNA for Prdxs are stored in SGs, which block their translation. In the later phase, the acute stress is perceived by the cell and induces the disassembly of SGs since transcription levels of mRNA binding proteins, such as TIAR and TTP, decrease so as to unlock of the translation for Prdxs.

## 5. Conclusions

Our data on the time course of the transcription levels and active proteins levels highlight that Cr-Prdx2, Cr-Prdx3, and Cr-Prdx4 perform their antioxidant function in an integrated way. Regarding Cd and Cu, Cr-Prdx2 is the most active isoform, between the three we studied, in contrasting acute stress induced by the metals. In the presence of Zn, we hypothesize a post-transcriptional control operated by stress granules. Our results, clarifying the importance of peroxiredoxins in protecting organisms from heavy metal pollution, are the first obtained in tunicates. Future studies will characterize and study, at the transcriptional and post-transcriptional levels, all the peroxiredoxin isoforms of *C. robusta*, the putative gene sequences present in the ANISEED database, in order to better understand how they work together in contrasting oxidative stress. In addition, a deeper insight into the role of Prdxs in *C. robusta* biology can provide data valid for their use as stress biomarkers in biomonitoring campaigns of the marine ecosystem, using *C. robusta* as a sentinel organism.

## Figures and Tables

**Figure 1 antioxidants-11-00093-f001:**
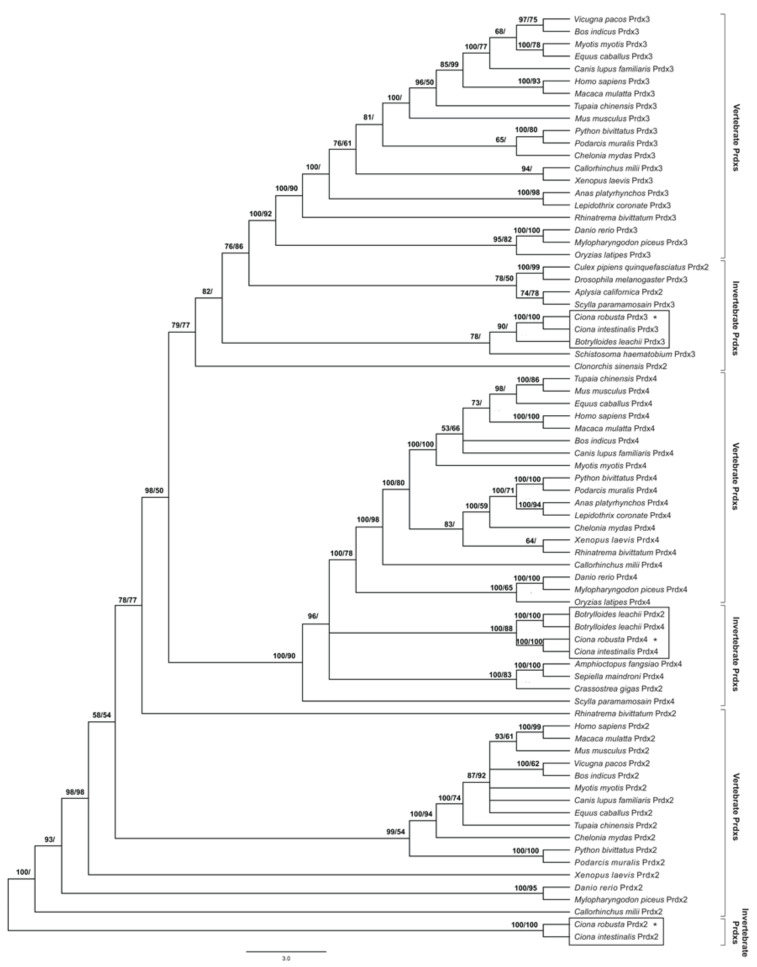
Phylogenetic relationships among Prdxs of various organisms reconstructed on the basis of the cDNA coding region sequences and using both Bayesian interference (BI) and maximum likelihood (ML) methods. Bayesian posterior probability (first number) and bootstrap values higher (and equal to) than 50% are indicated on each node, respectively. The scale for branch length (3.0 substitution/site) is shown below the tree. Asterisks indicate the C. robusta Prdxs.

**Figure 2 antioxidants-11-00093-f002:**
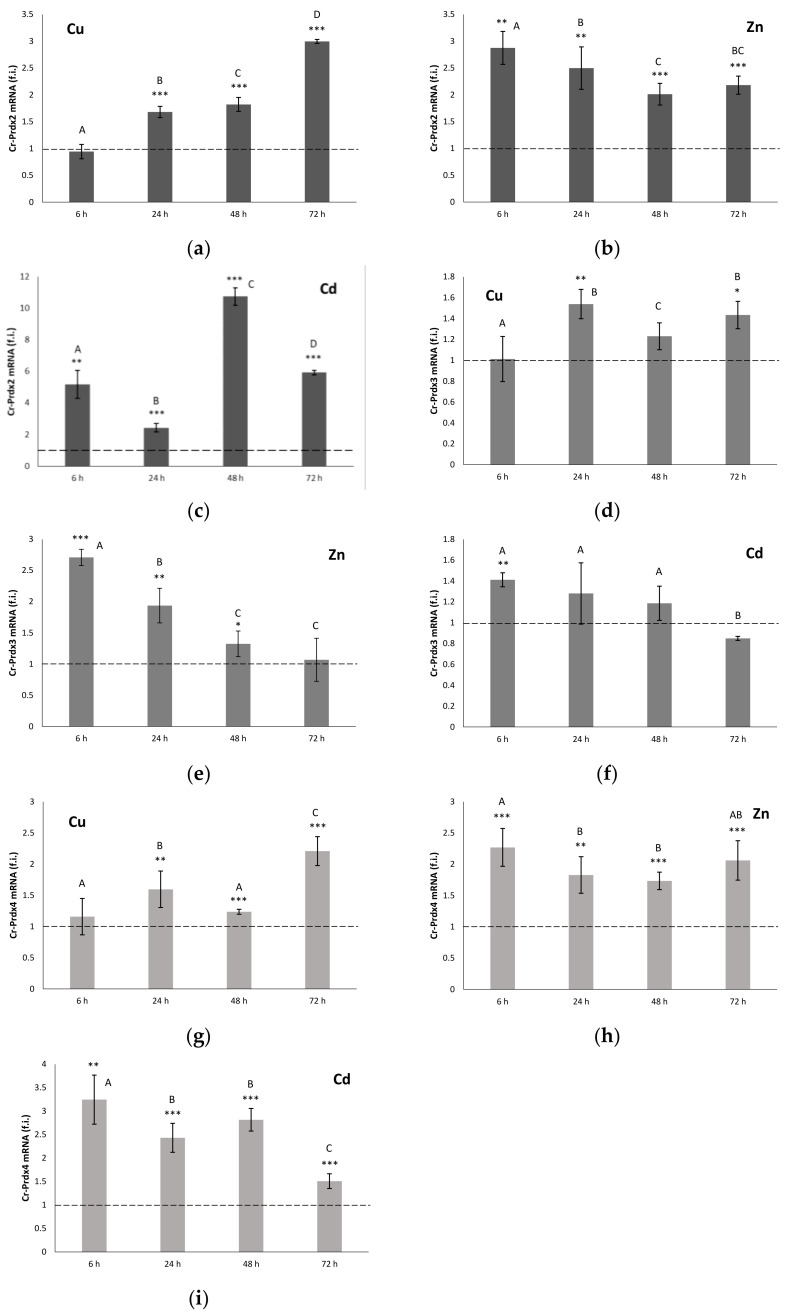
Relative expression levels (fold induction, f. i.) of Cr-Prdx2 (**a**-**c**), Cr-Prdx3 (**d**-**f**) and Cr-Prdx3 (**g**-**i**), during Cu, Zn and Cd exposure. Values are indicated as mean ± SD. Transcription levels were normalized with respect to controls (dashed line). Asterisks: significant differences with respect to controls (*** *p* < 0.001, ** *p* < 0.01, * *p* < 0.05). Different letters correspond to significant statistical differences (*p* < 0.05) among different treatment times (Duncan’s test).

**Figure 3 antioxidants-11-00093-f003:**
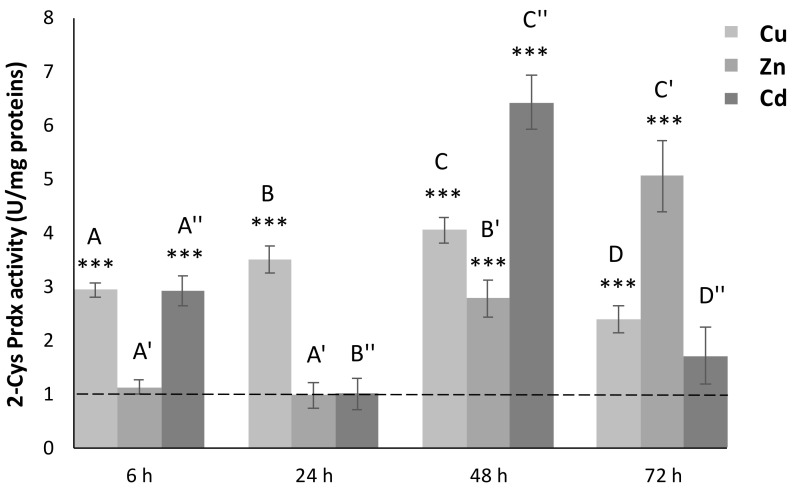
2-Cys Prdx activity levels in the intestine of *C. robusta* during metal exposure (Cu, Zn and Cd). Values are indicated as mean ± SD. Protein levels were normalized with respect to controls (dashed line). Asterisks: significant differences with respect to controls (*** *p* < 0.001). Different letters with the same index correspond to significant statistical differences (*p* < 0.05) among different treatment times (Duncan’s test).

**Figure 4 antioxidants-11-00093-f004:**
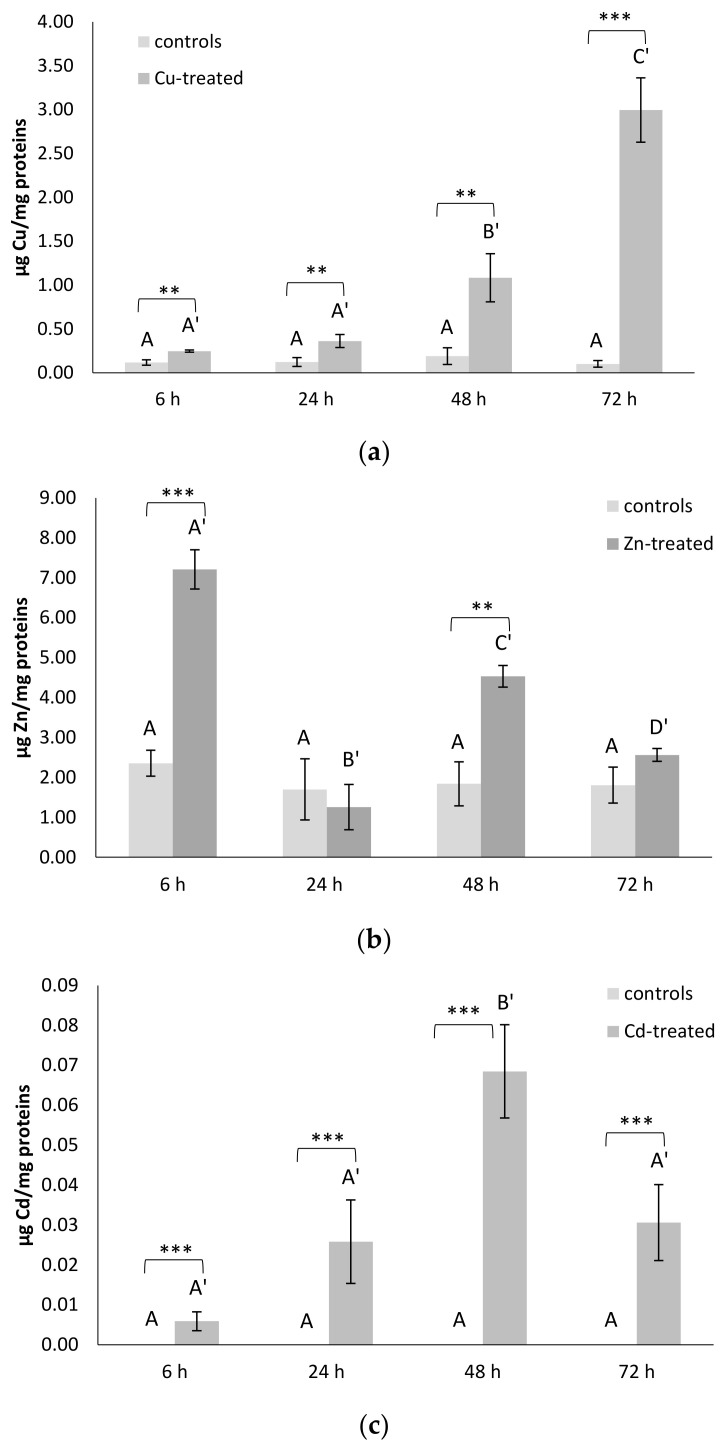
(**a**) Cu, (**b**) Zn and (**c**) Cd accumulation in the intestine of *C. robusta* (µg metal/mg total protein). Values are indicated as mean ± SD. Asterisks: significant differences with respect to controls (*** *p* < 0.001, ** *p* < 0.01). Different letters with the same index correspond to significant statistical differences (*p* < 0.05) among different treatment times (Duncan’s test).

**Figure 5 antioxidants-11-00093-f005:**
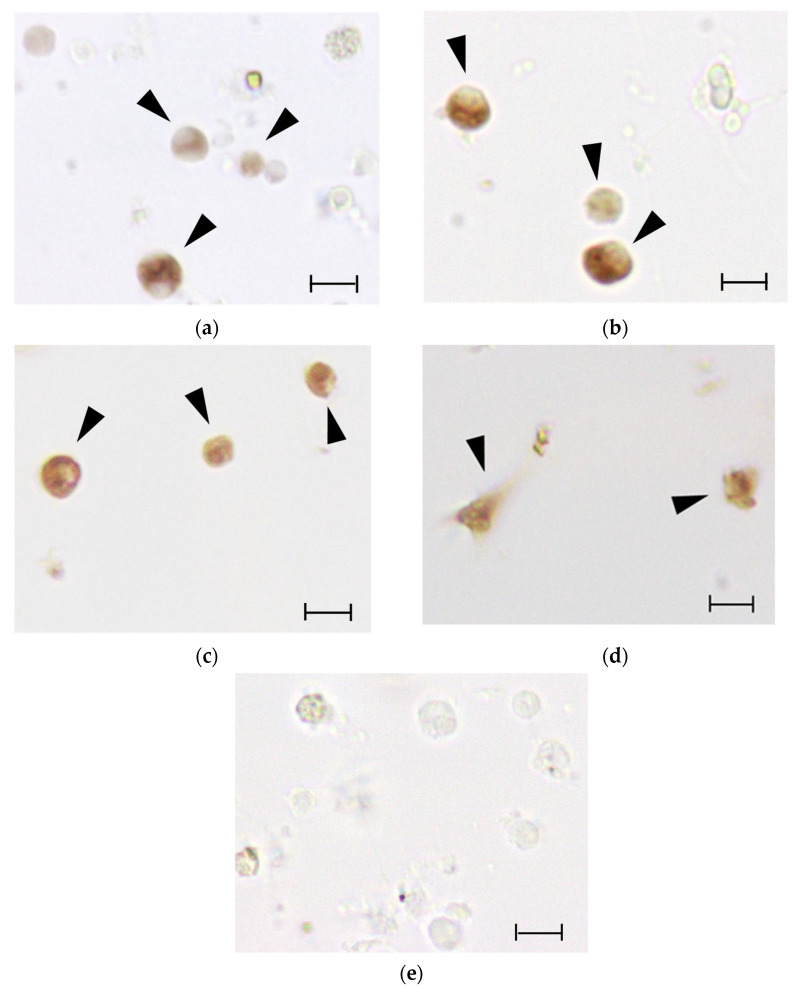
ISH with anti-sense riboprobes for (**a**) *cr-Prdx2*, (**b**) *cr-Prdx3* and (**c**,**d**) *cr-Prdx4*, and sense riboprobe, for (**e**) *cr-Prdx4* as reference, on hemocyte monolayers from three-day Cd treatment. Brown color: riboprobe staining. Arrowheads: labeled hemocytes. Scale bar: 10 µm.

## Data Availability

Data are contained within the article and Appendix A.

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
