# Peer review of "Typical 2-Cys Peroxiredoxins as a Defense Mechanism against Metal-Induced Oxidative Stress in the Solitary Ascidian Ciona robusta"

_antioxidants, 2021, doi:10.3390/antiox11010093_

Round 1

Reviewer 1 Report

The authors have clearly demonstrated that these peroxiredoxins are conserved to Ciona robusta and that their expression levels are variably positively correlated to the accumulation of the three metals studied. An interesting study to suggest that these organisms provide a marine window to monitor the effects of pollution of ecosystems.

Author Response

Thank you very much for your comments.

Reviewer 2 Report

The article reports on the identification and initial characterization of peroxiredoxins (Prx) in a species of marine invertebrates. This article is simply a description of Prxs identified in another organism and a phylogenetic analysis. There is a long and detailed description of the observed changes in the levels of the three peroxiredoxin orthologs identified in the sea creature, Ciona robusta. The only thing that has been demonstrated is changes in mRNA levels in response to metals as determined by RT-PCR and in situ hybridization, but there is little novelty in this description. The results simply indicate the induction of Prx expression in response to metals, but this does not reveal the functional significance of these enzymes. It cannot be concluded that animals were better protected from the toxicity of these metals, since no other studies, including effects on physiology, have been carried out. In addition, changes in levels do not necessarily indicate a protective function that could have taken place and, in fact, has been amply demonstrated in experiments with various mutants and transgenic overexpressors in various model organisms, both mammals and invertebrates.

Discussion is more of a review of known published data rather than an analysis and comparison of the results obtained in this study.

There are problems with experimental design:

  1. The authors report that the experiments used six animals per one time point, and I assume that this was only one biological repeat. Experiments must be performed with at least two independent biological replicates.
  2. Determination of enzymatic activity cannot be achieved in crude cell lysates, since there is no point in measuring the peroxiredoxin enzymatic activity in a complex mixture of macromolecules and low-molecular compounds that can interfere with the reaction.
  3. The peroxiredoxin activity measurement method using NADPH and thioredoxins used by the authors was actually developed for in vitro activity determination and was carried out with purified enzymes as reported in many published studies. This method is not applicable to crude cell lysates.

Minor: Figure captions do not indicate how many measurements were taken and other important details are missing. For example, what is represented by the letter A, B, etc.?

Author Response

Thank you very much for your comments, but we would like to specify that the description and the novelties include:

(1) the molecular characterization of 3 Prdx isoforms for the first time in a cordate invertebrate (tunicates are the sister group of vertebrates) and appropriate in silico analysis (description of gene sequences and protein domain composition);

(2) the phylogenetic analysis applying the two most important methods for trees reconstruction of (ML and Bayesian), using both aa and nucleotide sequences and the study on negative selection, that allowed to increase the knowledgment of the molecular evolution of this enzyme;

(3) the gene expression analysis at pre- (qRT-PCR) and post- (tissue enzymatic activity) levels, in response to three different metal ions (both essential and non-essential), used as inducers of ROS production, that allowed  to formulate for the first time the hypothesis about the post-transcriptional control operated by stress granules;

These analyses were complemented by the evaluation of metal accumulation in intestine and by the in situ hybridization, which allowed to underestimate which are the haemocytes (immunocytes) involved in the expression of transcripts. I think that this in enough, also because other two Reviewer have the same opinion.

The obtained results clearly indicate that the three Prdx isoforms play their role of cell peroxide scavenger that is time- and metal- specific. We agree that other studies are needed to confirm the protection role of Prdxs. However, it is to note that these enzymes act together with other proteins to protect the cell from the oxidative stress risk, also induced by Cu, Zn and Cd accumulation. Therefore, we can assume that also in C. robusta the Prdxs induction represents a protective physiological response.  Unfortunately, mutants and transgenic animals are not physiological systems, and the information obtained from these experimental conditions can not be comparable to wildlife.

For the discussion, we amply referred to our and other previous papers because they focused on metal toxicity and antioxidant defences. Thanks to these references, we compared and integrated information in order to better understand the antioxidant responses of C. robusta. Obviously, more experiments and analyses must be performed obtain a reliable picture of this very complicate physiological defence system

Three more papers based on the identical experimental plan have been published on top rank journals. In addition, a very long list of papers reported experiments with only one repeat. The six animal used in our experiment were not pooled, so each specimens represent a biological repeat.

The enzymatic activity can be analysed in cell-free extract. Also in this case there is a very long list of papers reporting enzymatic activity analysis in cells, tissues and full organisms. Obviously, there is the possibility that some molecules can interfere with the assay, but considering the very high specificity of the method for the detection of active 2-Cys Prdxs, we assumed that the bias is negligible.

Figure captions, were implemented.

Reviewer 3 Report

This study will provide new aspect about Prx study in marine organisms and their response of antioxidant enzymes to heavy metals. I think this is a study equivalent to acceptance of this journal. The details are as follows.

Line 186(Statistical analysis in method)

In method, authors mentioned that Duncan's test was used. But in each figure caption, there is no description about Duncan's test and all test methods in figure captions are Student-Newman-Keuls t-comparison.

Line 404-405

Prdx3level at 72h is almost same with control not consistently higher.

Figure1-3

It is hard to compare among graphs due to that graph size is too big. Please show 9 graphs within 1 page.

In addition, please change comma , to dot . in decimal point value of Y axis values (Fig6 also).

Line 629-630

In the Prdx3, there was no remarkable difference of expression level between Cd and Zn.

Author Response

Thank you very much for your comments.

In figure captions “Student-Newman-Keuls t-comparison“ changed in “Duncan's test”.

“consistently higher than the controls in the organisms exposed to Zn“ changed in “consistently higher than the controls in the organisms exposed to Zn for the first 48 h“.

We included the 9 graphs in a single page and figure.

“Exposure to Zn also has much less marked effects on the transcription of genes encoding Cr-Prdx isoforms 2, 3, and 4 than Cd” changed in “Exposure to Zn also has much less marked effects on the transcription of genes encoding Cr-Prdx isoforms 2 and 4 than Cd”